# Detection of Chrysanthemums Inflorescence Based on Improved CR-YOLOv5s Algorithm

**DOI:** 10.3390/s23094234

**Published:** 2023-04-24

**Authors:** Wentao Zhao, Dasheng Wu, Xinyu Zheng

**Affiliations:** 1College of Mathematics and Computer Science, Zhejiang A&F University, Hangzhou 311300, China; 2Key Laboratory of State Forestry and Grassland Administration on Forestry Sensing Technology and Intelligent Equipment, Hangzhou 311300, China; 3Key Laboratory of Forestry Intelligent Monitoring and Information Technology of Zhejiang Province, Hangzhou 311300, China

**Keywords:** target detection, flowering recognition, YOLOv5, coordinate attention mechanism, structural reparameterization

## Abstract

Accurate recognition of the flowering stage is a prerequisite for flower yield estimation. In order to improve the recognition accuracy based on the complex image background, such as flowers partially covered by leaves and flowers with insignificant differences in various fluorescence, this paper proposed an improved CR-YOLOv5s to recognize flower buds and blooms for chrysanthemums by emphasizing feature representation through an attention mechanism. The coordinate attention mechanism module has been introduced to the backbone of the YOLOv5s so that the network can pay more attention to chrysanthemum flowers, thereby improving detection accuracy and robustness. Specifically, we replaced the convolution blocks in the backbone network of YOLOv5s with the convolution blocks from the RepVGG block structure to improve the feature representation ability of YOLOv5s through a multi-branch structure, further improving the accuracy and robustness of detection. The results showed that the average accuracy of the improved CR-YOLOv5s was as high as 93.9%, which is 4.5% better than that of normal YOLOv5s. This research provides the basis for the automatic picking and grading of flowers, as well as a decision-making basis for estimating flower yield.

## 1. Introduction

Fresh-cut flowers are mainly sold in pre-order mode, and the transportation distance is strongly correlated with the picking time (for short-distance transportation with picking flowers in full bloom, and for long-distance transportation with picking flowers in bud), which requires a more accurate estimation of various flowering stages of flowers. Conventional manual estimation methods of flower yield are inefficient, time-consuming, and low-accuracy, leading to difficulties in timely order delivery. Flowering stage recognition is essential for automated grading and production yield estimation. Therefore, applying computer vision technologies based on deep learning algorithms to monitor the flowering stage changes of chrysanthemums allows enterprises to take timely intervention measures to guarantee production and sales.

Various flower classification methods based on deep learning have been widely used in recent years to recognize flowers from their background, such as leaves. Significantly, the convolutional neural networks (CNN) containing multilayer stacked structures have obtained better recognition results in recognizing flower color, shape, and appearance features [1,2]. Dias et al. [3] predicted the flowering intensity of apple blossoms by fine-tuning a pre-trained convolutional neural network (CNN) and obtained an estimation accuracy and recall of more than 90%, however, at the cost of a slower detection speed. Oppenheim et al. [4] proposed a greenhouse tomato detection algorithm based on machine vision, which was used to count the yellow tomato flowers under complex conditions such as different growth environments, illumination, and flower sizes. Although this algorithm successfully identified the blooming yellow flowers of tomatoes using an unmanned aerial vehicle (UAV) at close range in the greenhouse, it does not classify various flowering periods. Cıbuk et al. [5] proposed a novel method for flower species classification that combined the concatenated AlexNet and VGG-16 models, the mRMR feature selection technique, and the SVM classifier with an RBF kernel. They achieved remarkable results of 96.39% and 95.70% accuracy on the Flower17 and Flower102 datasets, respectively. Tian et al. [6] proposed a deep learning algorithm to detect and recognize flowers by introducing a Single Shot MultiBox Detector (SSD), which has been evaluated on the flower dataset published by Oxford University and achieved an average precision of 83.64% on VOC2007 and 87.4% on VOC2012. The method also incorporates a video stream scheme to address the detection of two or more target flowers. Feng et al. [7] have proposed a recognition model to distinguish different flower categories based on the VGG16 model and the Adma (Adaptive Moment Estimation) deep learning optimization algorithm. This approach harnesses the power of transfer learning to expedite network convergence and has demonstrated a remarkable recognition accuracy of 98.99% on a dataset consisting of 30 distinct flower categories. Lawal M O. et al. [8] proposed an improved YOLOv3 model to detect tomatoes under complex environmental conditions by applying dense structure merging, the Mish activation function, and spatial pyramid pooling. Through assessing the impact of various input layers on the network’s performance and refining the pruning strategy by comparing the γ coefficients of the Batch Normalization (BN) layers of the trained YOLOv4 model, Wu et al. [9] used a channel pruning-based YOLO v4 algorithm to detect apple blossoms in natural environments and achieved a faster detection speed. Farjon et al. [10] employed Faster-RCNN and incorporated professional growers’ labeling of flower information to distinguish different flowering levels of canopy apple blossoms. The proposed model achieved an average accuracy of 68% in discriminating and classifying apple flowers with varying opening degrees. Li et al. [11] enhanced the YOLOv5 algorithm’s detection accuracy of pears by integrating a Transformer Encoder, demonstrating a maximum average accuracy of 96.12% and robustness improvement in different shading and lighting conditions. After YOLOv5, new algorithms such as YOLOv6 [12], YOLOv7 [13], YOLOv8 [14], YOLOX [15], and YOLOF [16] have been proposed in recent years.

A combination of computer vision techniques and deep learning algorithms have been extensively employed in various fields of agricultural production, and numerous studies have demonstrated promising outcomes [17,18,19]. However, most existing studies on flower recognition only focused on identifying flower species and not flowering period recognition [20,21,22]. This study aims to find a lightweight and high-precision YOLO algorithm and attempts to improve it to be more suitable for use on inexpensive camera devices to identify and distinguish the flowering periods of yellow chrysanthemums.

## 2. Materials and Methods

### 2.1. Data Collection

The original dataset includes bud and bloom images of chrysanthemums taken by camera equipment in a chrysanthemum plantation. The greenhouse is about 3 m high, and chrysanthemums are about 50–70 cm above the ground. Agricultural facilities such as lighting and sprinklers are generally installed at about 2.5 m. The camera needs to be a certain distance above the chrysanthemum to achieve effective collection. Therefore, we collected data at a height of 2–2.5 m. The dataset was manually labeled with the aid of LabelImg, and 250 images were selected after data cleaning. The dataset is split into a training-validation set and test set by a ratio of 9:1, and the training-validation set is further split by a ratio of 9:1. Figure 1 illustrates sample images of flower buds (Figure 1b) and blooming flowers (Figure 1a). Correspondingly, the number of flower buds and blooming flowers are shown in Table 1.

### 2.2. YOLOv5 Model

The YOLOv5 detection algorithm released by Ultralytics in 2020 is a lightweight model. YOLOv5 utilizes two parameters of multiple-depth (named depth_multiple) and multiple-width (named width_multiple) to control four structures: YOLOv5s, YOLOv5m, YOLOv5l, and YOLOv5x. YOLOv5s consists of three parts: backbone, neck, and head, as shown in Figure 2.

The Yolov5s detection algorithm utilizes CSPDarknet as the backbone feature extraction network, encompassing four crucial characteristics: Residual, CSPnet, SiLU activation function, and Spatial Pyramid Pooling—Fast (SPPF) structure.

The Residual facilitates the optimization and accuracy enhancement of the model by adding significant depth. Additionally, the internal residual blocks employ jump connections to counteract the gradient disappearance that arises with increasing depth in deep neural networks.

The CSPnet network structure, generated from the CSPNet design idea, splits the input into two branches for convolution operation and some residual blocks and then splices the two branches to reduce computation while maintaining accuracy.

SiLU (Figure 3) is an improved function based on Sigmoid and ReLU. It is non-monotonic, smooth, and without upper or lower bounds.

The SPPF structure evolved from Spatial Pyramid Pooling (SPP) to reduce computational workload and enhance network speed. Its feature extraction is achieved through maximum pooling with different pooling kernel sizes to improve the perceptual field of the network.

### 2.3. Coordinate Attention

Squeeze-and-Excitation attention (SE) is sufficient for modeling. However, it needs to pay more attention to the importance of location information, which is crucial for explaining the spatial structure of visual targets [23]. The Convolutional Block Attention Module (CBAM) incorporates location information by globally pooling over channels to address this issue. However, this approach can only capture local information and obtain short-range dependent information [24]. Specifically, CBAM uses a weighting scheme that takes the maximum and average values of multiple channels at each location. As a result, this approach only considers local information but lacks global context information; therefore, it can’t capture the spatial structure of the targets well.

As shown in Figure 4, the coordinate attention (CoordAtt) [25] mechanism provides a novel and efficient approach for attention mechanisms by embedding location information into channel attention. This approach allows for acquiring more significant scale information with less overhead. The mechanism replaces the 2-D global pooling with two parallel 1-D feature encodings, thereby preserving location information and reducing the calculational workload. The feature encodings integrate spatial coordinate information by aggregating input features into perceptual feature mappings in both vertical and horizontal directions. These feature mappings with orientation-specific information are further encoded into two separate attention mappings. Each of these attention mappings captures the long-range dependencies of the input feature maps along a spatial direction. This encoding approach ensures the location information is retained in the generated attention mappings. Subsequently, the two attention mappings are multiplied and applied to the input feature map to highlight relevant features. Specifically, for a given input X, each channel is encoded along the horizontal and vertical coordinates using two spatially-scoped pooling kernels of size (H, 1) or (1,W), respectively. Thus, the output of the c-th channel at height h can be formulated as:(1)$$z_{c}^{h}\left(h\right)=\frac{1}{W}\sum_{0\leq i\leq W}x_{c}\left(h,i\right)$$

Similarly, the output of the c-th channel of width w can be written as:(2)$$z_{c}^{w}\left(w\right)=\frac{1}{H}\sum_{0\leq j\leq H}x_{c}\left(j,w\right)$$

The above two equations aggregate features along two spatial directions to generate a pair of directional perceptual feature mappings. Afterward, the aggregated feature mappings generated by Equations (1) and (2) are stitched together and passed through the shared 1 × *1 convolutional (Conv) transform function $F_{1}$ to obtain:(3)$$f=\delta \left(F_{1}\left(\left[z^{h},z^{w}\right]\right)\right)$$

Here, f ($f\in R^{C/r\times \left(H+W\right)})\;\mathrm{is}$ an intermediate feature map that encodes the spatial information in the horizontal and vertical directions, $\left[z^{h},z^{w}\right]$ represents the concatenation of feature maps generated along the two spatial dimensions, and δ is a non-linear activation function. Subsequently, f is partitioned into two independent tensors, namely, $f^{h}\in R^{C/r\times H}$ and $f^{w}\in R^{c/r\times W}$, corresponding to the two spatial dimensions, respectively. These tensors are then fed into two 1/times1 convolutional transforms $F_{h}$ and $F_{w}$, to produce intermediate feature maps $g^{h}$ (as shown in Equation (4)) and $g^{w}$ (as shown in Equation (5)), both of which have the same number of channels as the input tensor X.
(4)$$g^{h}=\sigma \left(F_{h}\left(f^{h}\right)\right)$$
(5)$$g^{w}=\sigma \left(F_{w}\left(f^{w}\right)\right)$$

Outputs $\;g^{h}$ and $\;g^{w}$ are expanded and used as attention weights, respectively. The expanded attention weights are then used to modulate the original feature map X to produce the output Y of the proposed coordinate attention mechanism, as defined in Equation (6).
(6)$$y_{c}\left(i,j\right)=x_{c}\left(i,j\right)\times g_{c}^{h}\left(i\right)\times g_{c}^{w}\left(j\right)$$

The backbone network structure after adding the coordinate attention block is shown in Figure 5.

### 2.4. RepVGG Block

Enhancing the representation capability of the model is crucial in object detection. While convolutional neural networks (CNNs) have evolved rapidly and exhibited powerful multi-scale representation capabilities, they also tend to result in more extensive and slower models. Yolov5 utilizes a bottleneck structure to more effectively propagate gradient information. However, this also increases the computational complexity. To alleviate this issue, Xiaohan Ding et al. [26] proposed the RepVGG network, which employs the structural reparameterization technique to convert the multi-branch model into a single-branch model during the inference period. Thus, this paper introduces the RepVGG block to replace the convolution block of the YOLOv5s algorithm to improve the network’s learning ability. In this paper, the RepVGG block is used for down-sampling (stride = 2) during training (Figure 6).

The parallel multi-branch structure of the RepVGG block enhances the model’s representation capability. In the backbone network, the RepVGG block structure adopts a down-sampling strategy with a stride of 2 and reconstructs the 3 × 3 convolution of YOLOv5s by introducing a 1 × 1 convolution branch. Figure 7 shows the overall backbone network structure after integrating the coordinate attention and RepVGG blocks.

### 2.5. Structural Reparameterization

Combining the stronger representability of multi-branch models and the faster speed and more efficient memory of single-branch models, a structural reparameterization was designed by using RepVGG block (a ResNet-style multi-branch model) for training and converting to a VGG-style single-way model for inference. As shown in Figure 8, Figure 8a represents the network structure used in RepVGG training, and Figure 8b represents the network structure used in inference.

As illustrated in Figure 9, structural reparameterization involves two primary stages. In the first stage, the conv layer and BN layer are merged into a single 3 × 3 conv, and the branch containing only BN is also converted into a separate 3 × 3 conv. In the second stage, the 3 × 3 convolutional layers on each branch are merged into a single layer. This fusion technique brings more efficient computation and faster convergence during the training process, which is particularly important for large-scale deep-learning applications. By combining these linear operations, deep learning models can achieve better performance with less computational and memory requirements.

#### 2.5.1. Convergence of Conv and BN

In deep learning architectures, convolution (Conv) and batch normalization (BN) layers are commonly used to extract features and accelerate training convergence. These layers are typically implemented as distinct operations, but their linear characteristics allow them to be fused into a single operator to improve computational efficiency and reduce memory overhead. The number of channels in each convolutional kernel is equal to the number of channels in the input feature map in the Conv layer, and the number of kernels determines the number of channels in the output feature map. The inference time of the BN layer consists of four main parameters: accumulated mean (μ), variance (σ^2^), learned scaling factor (γ), and bias (β). Here, the values of μ and σ^2^ are obtained by statistics, while γ and β are obtained through learning during the training process.

The calculation formula for the i-th channel of the feature map in the BN layer is expressed by Equation (7). Here, ϵ is a minimal constant as a stabilizer to prevent dividing by zero.
(7)$$y_{i}=\frac{x_{i}-\mu_{i}}{\sqrt{\sigma_{i}^{2}+\epsilon }}\cdot \gamma_{i}+\beta_{i}$$

Equation (8) is reformulated from Equation (7). Here, M represents the input feature map (Activation) of the BN layer, and ϵ is omitted for simplicity.
(8)$$bn{\left(M,\;\mu ,\;\sigma ,\;\gamma ,\;\beta \right)}_{:,\;i,:,:}\;=\;\left(M_{:,\;i,:,:}-\mu_{i}\right)\;\frac{\gamma_{i}}{\sigma_{i}}+\beta_{i}$$

The updated calculation formulas for the convolutional layer’s weight parameters are expressed as Equations (9) and (10). Here, W′ and b′ represent the updated weights and biases for the i-th convolutional kernel, respectively.
(9)$$W_{i,:,,,:}^{\prime }=\frac{\gamma_{i}}{\sigma_{i}}W_{i,,,,,,:}$$
(10)$$b_{i}^{\prime }=\beta_{i}-\frac{\mu_{i}\gamma_{i}}{\sigma_{i}}$$

#### 2.5.2. Converting 1 × 1 Layer to 3 × 3 Layer

Figure 10 illustrates the convolutional kernel within the 1 × 1 convolutional layer. By enhancing the original weights with a circular border of zeros, a 3 × 3 convolutional layer is obtained. It is worth noting that to ensure that the height and width of the input and output feature maps remain unchanged, the padding must be set to 1 (the original convolutional kernel size is 1 × 1 with the padding of 0). Finally, the convolution and BN layers can be fused following the steps described in Section 2.5.1.

#### 2.5.3. Multi-Branch Fusion

After performing the fusion of Conv and BN, as well as the conversion of 1 × 1 layer to 3 × 3 layer, the entire convolution block is transformed into a multi-branch structure composed entirely of 3 × 3 convolutions. According to Equation (10), the resulting structure can be converted into a single 3 × 3 convolution layer (as shown in Equation (11)) by simply adding the parameters to each convolutional layer. Here, ⊗ denotes the convolution operation.
(11)$$O=\left(I⊗K_{1}+B_{1}\right)+\left(I⊗K_{2}+B_{2}\right)+\left(I⊗K_{3}+B_{3}\right)=I⊗\left(K_{1}+K_{2}+K_{3}\right)+\left(B_{1}+B_{2}+B_{3}\right)$$

## 3. Results and Discussion

### 3.1. Experimental Environment Configuration

The hardware environment used for the network training in this paper is a Tencent Cloud server with Intel Xeon Cascade Lake 8255C (2.5 GHz), 10-core vCPU, 40 GB RAM, and 1 Tesla V100-NVLINK-32. The software environment is a combination of Ubuntu20.04 + python3.8 + pytorch1.10.2.

The size of the images sent to the network for training is uniformly set to pixels of 640 × 640, and the batch size is set to 32. To increase the diversity of training samples and improve the robustness, Mosaic data augmentation is used to train 300 epochs.

### 3.2. Evaluation Indicators

The classifier for the measured case and predicted case in the experiment is shown in Table 2. True Positive (TP) means both the measured case and the predicted case are positive. False Positive (FP) means the measured case is negative, and the predicted case is positive. False Negative (FN) means the measured case is positive, and the predicted case is negative. True Negative (TN) means both the measured case and the predicted case are negative.

This study employs a series of performance indicators to evaluate the efficacy of the proposed approach. These indicators include precision (P), recall (R), and mean average precision (mAP). P refers to the ratio of correctly identified positive samples to the total number of predicted positive samples. R corresponds to the ratio of correctly identified true positive samples to the overall number of measured positive samples. The average precision (AP) is the average of the precision values over the area under the Precision-Recall curve and the coordinate axes and is typically computed by integration methods. The mean average precision (mAP@.5) is the average of the AP values for each detected category. The mAP value is usually calculated using Intersection over Union (IoU) with a threshold of 0.5. Specifically, the indicators are determined by the following formulas.
(12)$$IoU=\frac{A\cap B}{A\cup B}$$
(13)$$P=\frac{TP}{TP+FP}$$
(14)$$R=\frac{TP}{TP+FN}$$
(15)$$AP=\int_{0}^{1}P\left(r\right)dr$$
(16)$$mAP=\frac{\sum_{i=1}^{S}AP_{i}}{S}$$

In Equation (12), A and B are the measured and predicted values, respectively. In Equation (16), S is the number of detected categories, $AP_{i}$ represents the accuracy rate of the i-th category.

### 3.3. Training Results

The improved model combines a modified attention mechanism and RepVGG block based on the same experimental configuration as the original YOLOv5s algorithm. Specifically, the improved algorithm is trained throughout 300 rounds using the generated training dataset. The outcomes of this training process are illustrated in Figure 11, where the blue line represents the average accuracy curve for an IoU threshold of 0.5, and the red line corresponds to the average accuracy curve for an IoU threshold between 0.5 and 0.95 with a step size of 0.05. The horizontal axis denotes the number of training rounds, while the vertical axis represents the mAP values. Obviously, the model exhibits slow learning ability before the 50th round of training, followed by a rapid convergence between rounds 50 and 150. Moreover, the mAP values gradually tend to stabilize after 200 training rounds.

### 3.4. Algorithm Performance Comparison

To compare the performance of the proposed CR-YOLOv5s algorithm to other algorithms of conventional YOLOv5s and typical deep learning models, we conducted a comparative experiment and listed the results in Table 3. Here, ten algorithms were involved in the experiment, that is, Faster R-CNN [27], SSD [28], YOLOv3-Efficient [29], YOLOv5s [30], YOLOv6 [12], YOLOv7 [13], YOLOv8 [14], YOLOX [15], YOLOF [16], and the improved CR-YOLOv5s. The results showed that the YOLOv5s algorithm significantly improved the mAP compared to the algorithms of Faster R-CNN and SSD. It is worth noting that considering the lightweight network in the YOLO series, YOLOv5s is superior to YOLOv3 in all performance indicators. In addition, the mAPs of YOLOv6 and YOLOv7 are also less than YOLOv5s. Admittedly, the mAP of YOLOX has slightly improved than that of YOLOv5s, however, the number of parameters significantly increased. Regarding the latest YOLO series algorithm YOLOF and YOLOv8, although YOLOv5s has a slightly lower mAP, its parameters are only 6.69 MB, far lower than the 10.61 MB of YOLOv8 and 42.31 MB of YOLOF. Some new blocks introduced by YOLOv8, such as C2f block used in backbone network and Neck. It optimizes the model structure by increasing gradient flow and adjusting the number of channels. However, it may cause more hardware overhead. Consequently, the YOLOv5s has been selected as the baseline for this paper.

To improve the classification accuracy for various flowering stages, even in complex environmental interferences, the coordinate attention block was introduced. Additionally, the RepVGG block was also used to enhance feature representation through a multi-branch structure. A comparison of performance indicators between the improved YOLOv5s and the original YOLOv5s revealed that the former significantly improves mAP and fps with fewer computational parameters.

### 3.5. Ablation Experiments

The following ablation experiments were designed to verify the effectiveness of the CoordAtt and RepVGG block. Based on YOLOv5s, the CoordAtt, RepVGG block, or both, were added to the experiments. Using the same training data set, the evaluation metrics included precision rate P, recall rate R, and mean average precision (mAP). The results of the ablation experiments are shown in Table 4.

The CoordAtt embedded location information into channel attention to allow the mobile networks to obtain larger-area information without increasing significant overheads. The result showed that adding the coordinate attention block increased mAP with a 2.2% improvement from 89.4% to 91.6%. It demonstrates that embedding location information into channel attention and increasing the weight values of different flowering periods with the CoordAtt mechanism can improve the detection accuracy for chrysanthemums.

According to the experimental results, replacing the convolution block in YOLOv5s with the RepVGG block increased the mAP of the model from 89.4% to 91.8%, a 2.4% improvement. This suggests that the RepVGG block enhances the modeling capability of the YOLOv5s and improves flower detection accuracy.

Compared to the original YOLOv5s, the incorporation of both the coordinate attention block and the RepVGG block led to a slight loss of recall (with a 1.1% reduction) and a significant increase in accuracy (14.0% increase in accuracy, and 4.5% increase in average accuracy). Furthermore, the best optimization was generated by integrating the coordinate attention mechanism and RepVGG block into YOLOv5s.

### 3.6. Comparison and Analysis of Algorithm Detection Results

Figure 12 illustrates the detection results between the C-YOLOv5s (adding coordinate attention block into the YOLOv5s) and the original YOLOv5s algorithm. Introducing the coordinate attention block could focus on more details for the images under a weak light environment, thus improving the detection capability and robustness of the network.

Figure 13 shows the detection results between the R-YOLOv5s (adding RepVGG block into the YOLOv5s) and the original YOLOv5s algorithm. Introducing the RepVGG block could improve the representation of the features and increase the perceptual field of network layers. Figure 13 also showed that the RepVGG block could detect targets with higher confidence, thus improving detection accuracy and speed.

Figure 14 shows the detection results comparing CR-YOLOv5s (adding CoordAtt and RepVGG block into YOLOv5s) and the original YOLOv5s. The experimental results revealed that introducing CoordAtt enables the network to focus on small objects in a larger area, enhancing the model’s robustness and detection accuracy. The introduction of the RepVGG block further enhanced the model’s representation ability during training by employing a multi-branch structure. The experimental results indicated that the proposed algorithm is the best combination of high detection accuracy and high detection speed.

## 4. Conclusions

In this paper, an improved algorithm CR-YOLOv5s is proposed to improve the detection accuracy for small targets and occluded targets in the recognition of bud and blooming flowers.

The coordinate attention mechanism is first introduced to enhance the receptive field and embed positional information into the channel attention, enabling the mobile network to obtain information from a larger area, thereby improving detection accuracy and robustness. In addition, the convolution blocks in YOLOv5s are replaced with the RepVGG block to enhance the representational capacity of the model at a lower cost.

The CR-YOLOv5s algorithm has better detection accuracy and can be well applied to flower stage detection. If more diverse and high-quality images can be obtained in the future, such as images with multiple shooting distances or flowering degrees, the proposed algorithm will be able to further improve the robustness and generality and may also be further extended to longer-term monitoring and subdivided flower stage recognition.

## Figures and Tables

**Figure 1 sensors-23-04234-f001:**
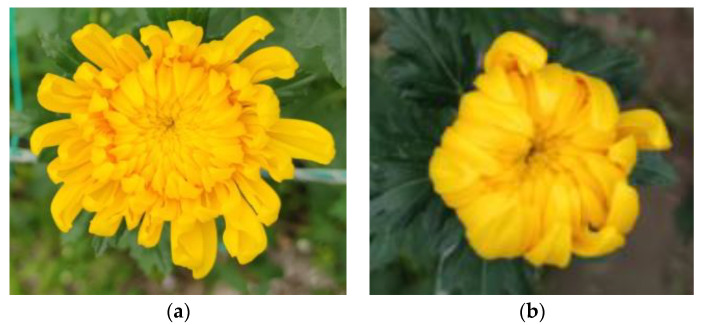
Labeling of yellow chrysanthemums in blooming and bud stage. (**a**) blooming stage. (**b**) bud stage.

**Figure 2 sensors-23-04234-f002:**
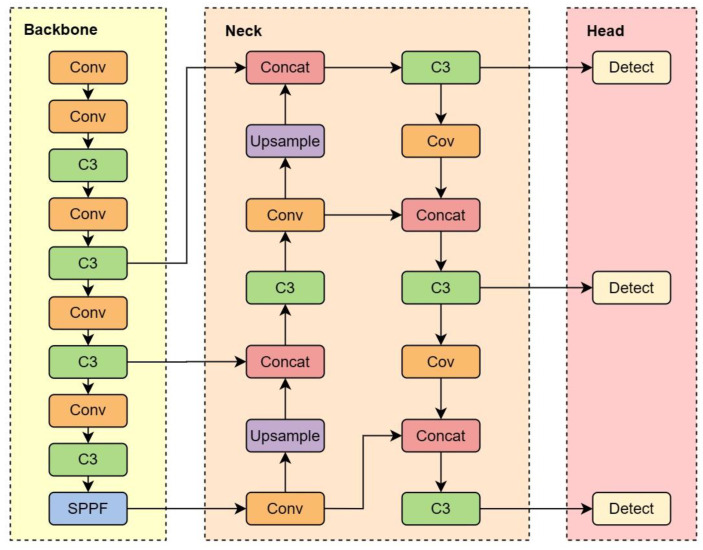
YOLOv5s structure diagram.

**Figure 3 sensors-23-04234-f003:**
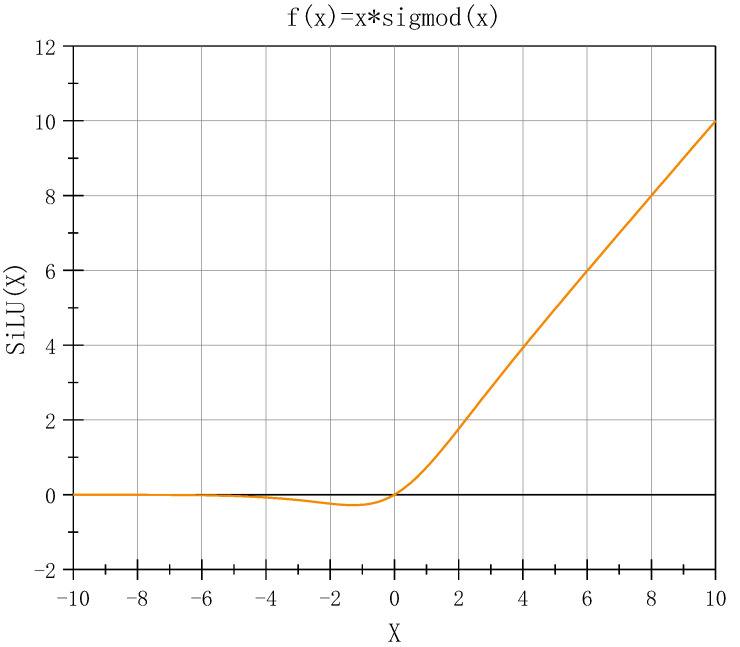
SiLU activation function.

**Figure 4 sensors-23-04234-f004:**
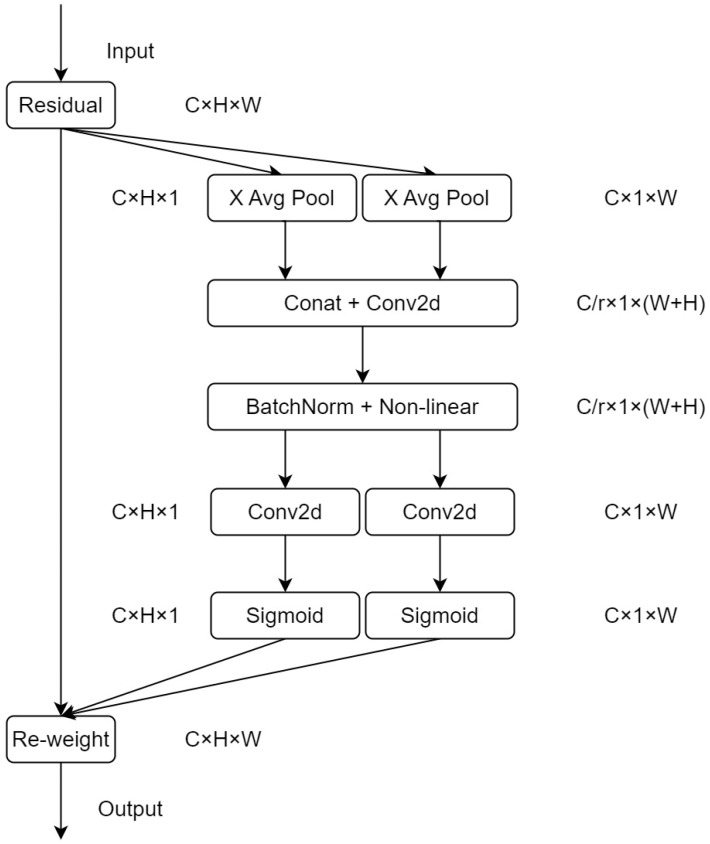
Coordinate attention (CoorAtt) block structure. Here, ‘X Avg Pool’ and ’Y Avg Pool’ refer to 1-D horizontal global pooling and 1-D vertical global pooling, respectively.

**Figure 5 sensors-23-04234-f005:**
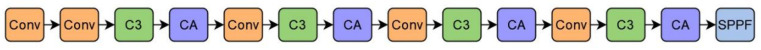
Backbone network structure after integrating coordinate attention block.

**Figure 6 sensors-23-04234-f006:**
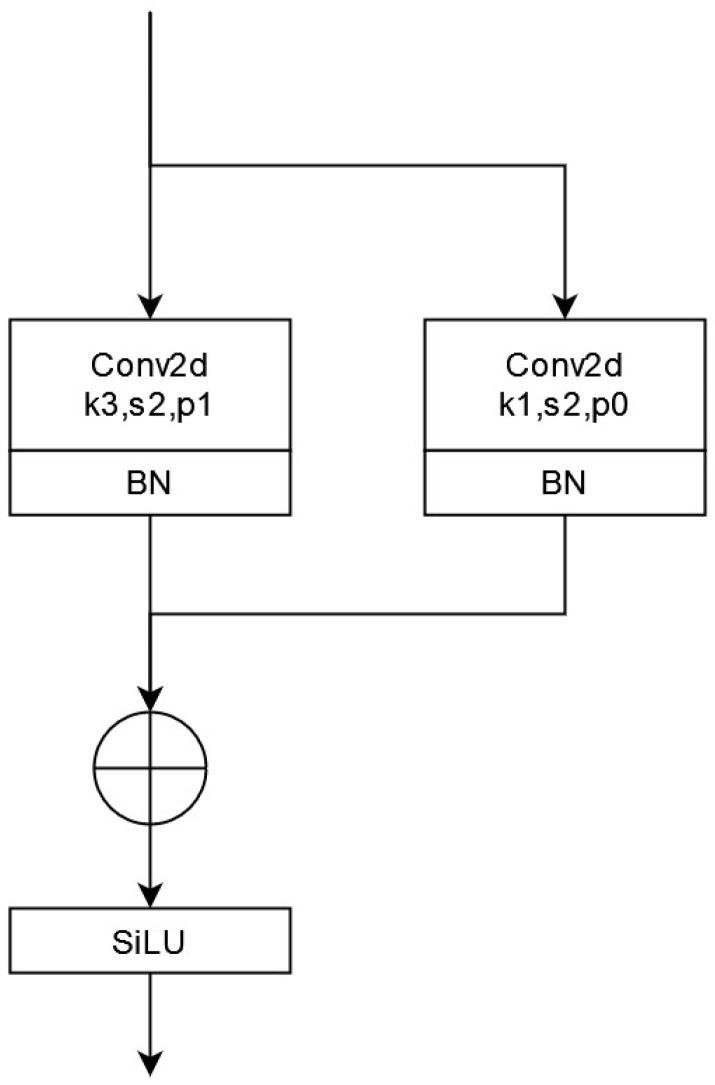
RepVGG block structure diagram.

**Figure 7 sensors-23-04234-f007:**
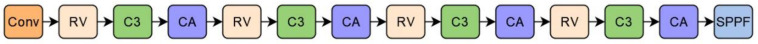
Structure of backbone network after integrating coordinate attention block and RepVGG block.

**Figure 8 sensors-23-04234-f008:**
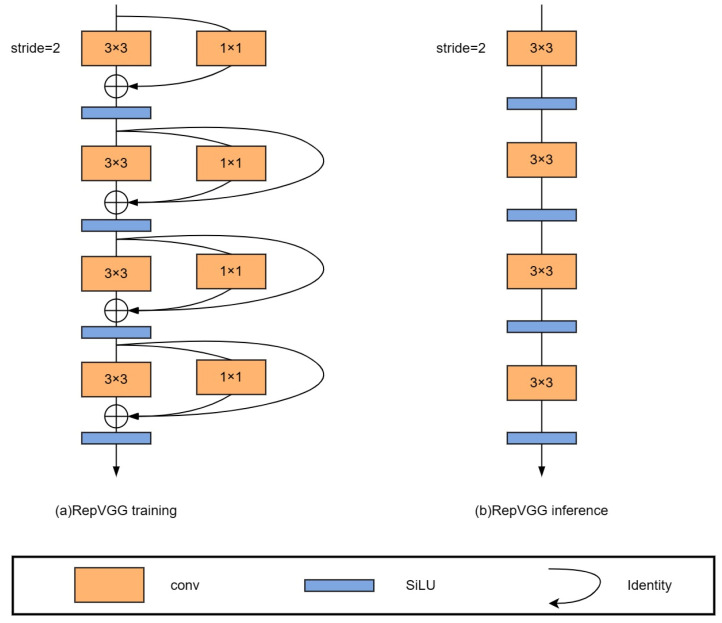
Sketch of RepVGG architecture.

**Figure 9 sensors-23-04234-f009:**
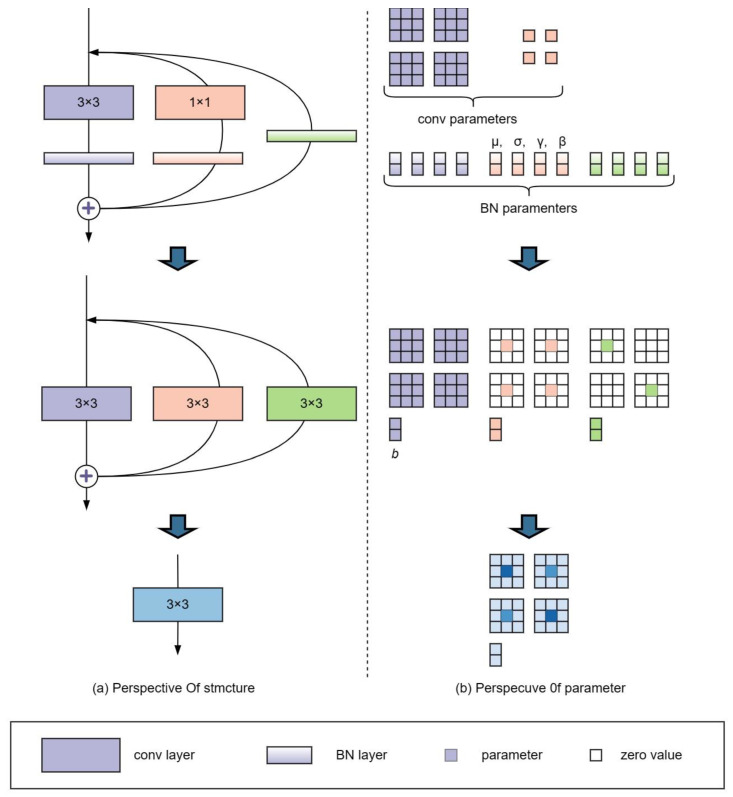
Structural reparameterization of RepVGG block.

**Figure 10 sensors-23-04234-f010:**
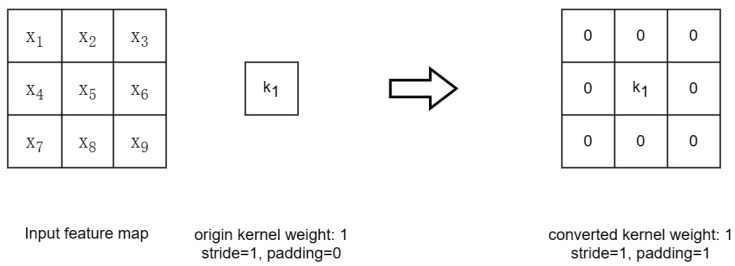
Converting 1 × 1 Conv to 3 × 3 Conv.

**Figure 11 sensors-23-04234-f011:**
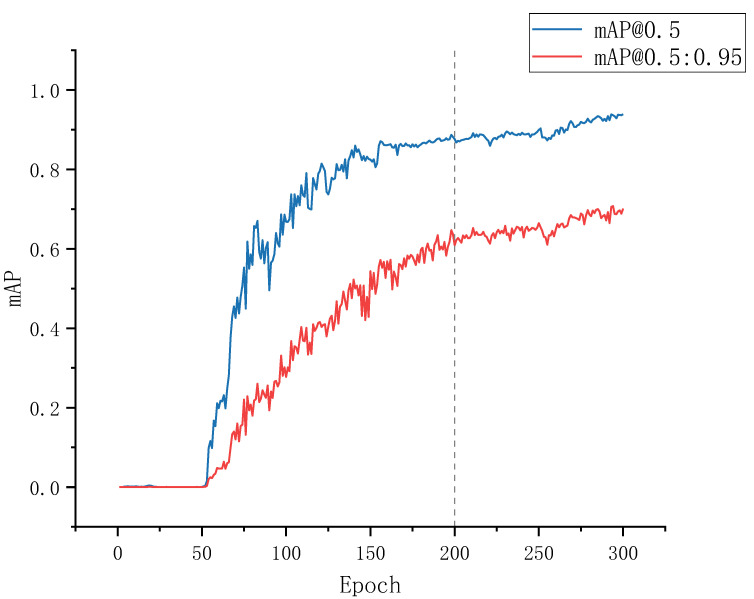
Model training epochs.

**Figure 12 sensors-23-04234-f012:**
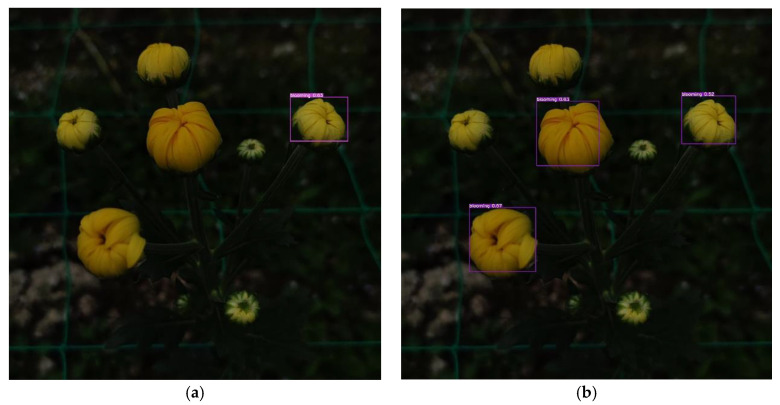
Detection results adding or not adding coordinate attention block. (**a**) YOLOv5s. (**b**) C-YOLOv5s.

**Figure 13 sensors-23-04234-f013:**
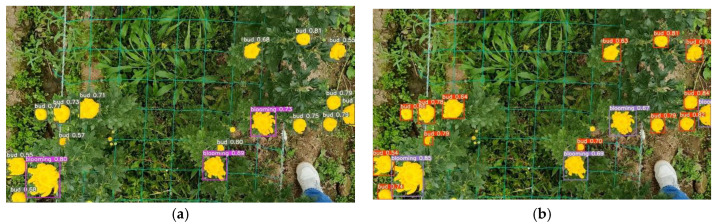
Detection results adding or not adding RepVGG block. (**a**) YOLOv5s. (**b**) R-YOLOv5.

**Figure 14 sensors-23-04234-f014:**
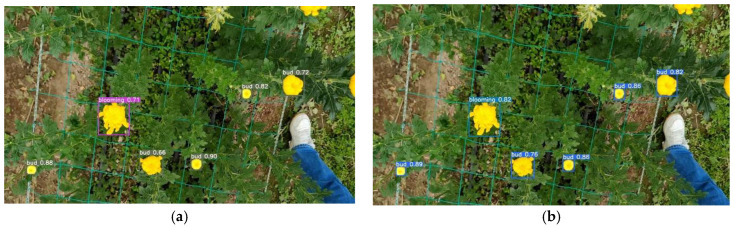
Detection results of adding or not adding both coordinate attention block and RepVGG block. (**a**) YOLOv5s. (**b**) CR-YOLOv5s.

**Table 1 sensors-23-04234-t001:** Structure of data set.

Objective	Number
bud	939
blooming	326

**Table 2 sensors-23-04234-t002:** Classification of measured and predicted samples.

Measured Case	Predicted Case	
	Positive Sample	Negative Sample
Positive sample	TP	FN
Negative sample	FP	TN

**Table 3 sensors-23-04234-t003:** Performance comparison.

Algorithm Model	Backbone	AP/%		mAP/%	fps	Params (MB)	GFLPs
		Bud	Blooming				
Faste R-CNN	Resnet50	79.34	87.81	83.58	10.35	137.1	370.21
SSD	VGG	78.32	86.66	82.49	23.11	26.29	62.747
YOLOv3	Efficientnet	86.64	73.45	80.04	63.49	7.22	4.042
YOLOv6	EfficientRep	91.50	86.00	88.80	73.13	4.63	2.83
YOLOv7	yolov7 backbone	91.00	77.00	84.10	34.66	37.62	106.472
YOLOv8	yolov8 backbone	89.00	94.50	91.80	135.14	10.61	28.4
YOLOX	Darknet53	89.30	98.74	89.52	95.24	54.21	156.011
YOLOF	ResNet	88.08	96.03	92.06	0.63	42.31	107.14
YOLOv5s-baseline	CSPDarknet53	94.50	84.4	89.40	24.75	6.69	15.8
ours	CSPDarknet53	94.40	93.4	93.9	19.08	6.4	15.2

**Table 4 sensors-23-04234-t004:** Ablation experiments.

Model	Precision/%	Recall/%	mAP_0.5/%
YOLOv5s	70.8	93.4	89.4
YOLOv5s + CoordAtt	84.4	88.1	91.6
YOLOv5s + RepVGG	86.7	82.1	91.8
YOLOv5s + CoordAtt + RepVGG	84.8	92.3	93.9

## Data Availability

No new data were created or analyzed in this study. Data sharing is not applicable to this article.

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
