# Peer review of "Detection of Chrysanthemums Inflorescence Based on Improved CR-YOLOv5s Algorithm"

_sensors, 2023, doi:10.3390/s23094234_

Round 1

Reviewer 1 Report

The manuscript proposed the flower recognition system with improved YOLOv5 as the baseline model. Here are the comments for the manuscript:

1. Why did the author not use YOLOv6 [1], YOLOv7 [2] or YOLOv8 [3] as the baseline model?

2. Please compare with the above 2020 methods, such as [1, 2, 3, 4, 5] and add the results in Table 3.

3. Please create another table for the processing time performance (Frame Per Second/FPS) to compare the proposed model with the model in Table 3.

References:

[1] Li, Chuyi, et al. "YOLOv6: A single-stage object detection framework for industrial applications." arXiv preprint arXiv:2209.02976 (2022).

[2] Wang, Chien-Yao, Alexey Bochkovskiy, and Hong-Yuan Mark Liao. "YOLOv7: Trainable bag-of-freebies sets new state-of-the-art for real-time object detectors." arXiv preprint arXiv:2207.02696 (2022). 

[3] https://github.com/ultralytics/ultralytics

[4] Chen, Qiang, et al. "You only look one-level feature." Proceedings of the IEEE/CVF conference on computer vision and pattern recognition. 2021.

[5] Ge, Zheng, et al. "Yolox: Exceeding yolo series in 2021." arXiv preprint arXiv:2107.08430 (2021).

Author Response

Dear Reviewer,

Reviewer 2 Report

The authors develop a variation on the YOLOv5 algorithm intended to characterize the growth state of cultivated flowers.   The proposed solution has a fundamentally practical interest. This is an article that does not try to delve into the frontier of knowledge in ML or computer vision paradigms, but addresses a practical problem in a productive sector of economic interest.   The article describes very precisely and with an excellent analysis of the proposed processing structure: it describes the basic proposal, analyzes its own problems (although it does not show them), and proposes the modifications that will result in better performance.   The formalism of the exhibition is really good and well explained.   The main problems come from the validation of the proposal, and are:   1 - the authors' proposal is compared with others... but they correspond to references that are almost 5 years old and that have not been used in the section dedicated to the state of the art to contextualize the proposal   2 - The intention of the paper is to develop a system that allows rationalizing and automating the process of collecting flowers for their transport and dispatch in the best conditions. However, the test features shown do not appear compatible with that goal.   3 - even when it is difficult, even sometimes impossible, to make an open and detailed study of what are the defining characteristics that a deep system learns, the evidence shown leaves the possibility that the only thing that the system is learning is the relative size of the sets of petals of the flowers, and not other characteristics of the flowering stage. this would make it difficult to use the system in the circumstances of a flower nursery, where the distances to the camera can be highly variable. Given the formulation of the problem, the latter two points constitute particularly serious ones.   Additional comments:   1 - line 36: incorrect acronym for deep learning 2 - lack of variability and representativeness in the references. The vast majority of them correspond only to Chinese researchers, which calls into question the representativeness and variability of the state-of-the-art described. 3 - a revision of the English used is recommended in order to correct minor errors of expression  

Author Response

Dear Reviewer,

Round 2

Reviewer 1 Report

Thank You very much for answering my comments. Based on the response letter, question number one is not answered yet. 

The question is: why not use the current YOLO version (i.e. YOLOv8) as the baseline and improve it with the proposed component (the convolution blocks from the RepVGG 19 block structure)? In Table 3, we can find that the YOLOv8 have much better performance in term of mAP and FPS rather than the YOLOv5s. Based on this information, why not choose the YOLOv8 as the baseline and improve it with the proposed component?

Author Response

Dear Reviewer,

Reviewer 2 Report

Corrections seem fine to me

Author Response

Dear Reviewer,

Thank you for your positive feedback. I appreciate your time and effort in reviewing my paper.

Round 3

Reviewer 1 Report

Thank you very much to answer my comment.

Good luck.